# Annual modulation of dark matter signals:
# Experimental results and new ideas

## Felix Kahlhoefer⋆

Institute for Theoretical Particle Physics (TTP),
Karlsruhe Institute of Technology (KIT), 76128 Karlsruhe, Germany

⋆ kahlhoefer@kit.edu

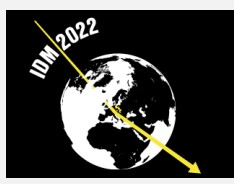

*14th International Conference on Identification of Dark Matter
Vienna, Austria, 18-22 July 2022*

## Abstract

Direct detection experiments searching for the scattering of dark matter particles off nuclei expect an annual modulation in their event rate. In this presentation, I will review the theoretical predictions and the experimental status of the search for annual modulations, with a focus on ongoing and planned experiments using NaI detectors. In particular, I will discuss the interpretation of the DAMA signal and related model-building efforts.

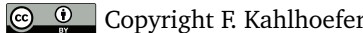

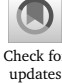

## 1 Introduction

One of the most intriguing predictions for the direct detection of dark matter (DM) in low-background underground experiments is that the event rate should exhibit an annual modulation. This modulation is a result of the rotation of the Earth around the Sun, which leads to a seasonal variation of the velocity of the Earth in the Galactic rest frame at the level of 10% (see Ref. [1] for details). As a result, both the overall flux of DM particles and their incoming velocity is predicted to peak around early June, which in turn shifts the spectrum of nuclear recoils from DM scattering to higher energies [2].

To first approximation, the time-dependence of the differential event rate can be written as

$$\frac{\mathrm{d}R}{\mathrm{d}E_\mathrm{R}} = \frac{\mathrm{d}\bar{R}}{\mathrm{d}E_\mathrm{R}} + \cos(\omega t + \phi)\frac{\mathrm{d}A}{\mathrm{d}E_\mathrm{R}}, \tag{1}$$

where the modulation amplitude $A(E_\mathrm{R})$ is predicted to switch sign at very low recoil energies (so-called anti-modulation) [3]. This variation of the DM signal with time can be exploited to

remove unknown but time-independent backgrounds and may serve as the ultimate proof for the DM origin of an observed signal.[1]

## 2 The elephant in the room

In fact, this situation is not hypothetical. For many years the DAMA collaboration has been observing an annual modulation in their experimental single-hit data, which exhibits a time dependence compatible with the expectations for DM scattering, and which has by now reached a significance of more than $13\sigma$ [8].

Interestingly, the question whether the energy dependence is also compatible with the expectations for DM scattering turns out to be more subtle. In the energy range 2–6 keVee it is indeed possible to fit the energy dependence of the modulation amplitude with various DM models, including both spin-independent and spin-dependent scattering on either iodine or sodium [9]. However, the DAMA collaboration has recently released new data with a low-energy threshold of only 0.75 keVee, which should make it possible to distinguish between these various possibilities [10]. For example, for spin-independent scattering on iodine, the anti-modulation should become visible at the lowest observable energies unless it is smeared out by the energy resolution. Preliminary results suggest that spin-independent scattering is strongly disfavoured (for the commonly assumed detector resolution and quenching factors), while spin-dependent scattering still gives an acceptable fit to data (see figure 1). Deeper insights can be expected if the threshold can be lowered even further.

Unfortunately, there is no official analysis of the DAMA data that would allow for a comparison with other direct detection experiments. Nevertheless, such comparisons have been performed using publicly available data, most recently in Refs. [11–15]. These studies show very clearly that for standard astrophysical assumptions the DAMA signal is incompatible with existing exclusion limits for any type of nuclear scattering. The case of elastic scattering can even be excluded independent of astrophysical assumptions.

This leads to the sobering conclusion that if the DAMA signal is in fact due to the scattering of DM particles, we must be fundamentally wrong about their astrophysical distribution and fundamental interactions. While this may seem unlikely, scientific progress requires that we find ways to independently test DAMA without the need for any such assumptions. In other words, we need to develop new NaI detectors capable of searching for annual modulations over the same energy range as DAMA. Unfortunately, this task is far from easy, as it remains challenging to achieve the necessary crystal purity and remove enough radioactive contaminants to produce detectors with the same ultra-low background rate as DAMA (of the order of 1 cpd/kg/keV in the region of interest).

## 3 Testing DAMA with NaI experiments

Nevertheless, two collaborations have recently achieved a level of background, and hence sensitivity, comparable to DAMA. The first is COSINE-100, which is a joint venture of KIMS and DM-Ice and operates at the Yangyang Underground Laboratory in South Korea. The total rate achieved by the experiment is low enough to exclude DAMA for standard assumptions [20],

---

[1]It is worth noting that many backgrounds are not time-independent. Indeed, the underground muon flux correlates with atmospheric temperature, such that one may suspect backgrounds from neutron scattering to exhibit a seasonal variation [4]. And even slowly decreasing backgrounds from intrinsic radioactivity may lead to an apparent modulation of the signal if the background subtraction is performed periodically [5,6], as illustrated very recently by the COSINE collaboration [7].

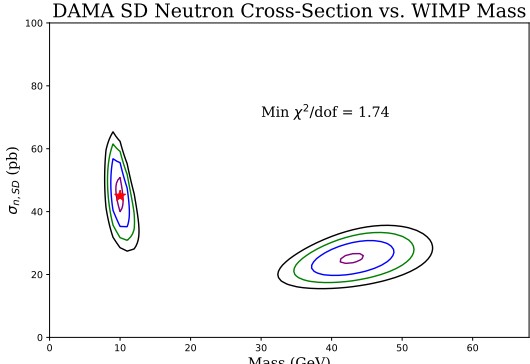
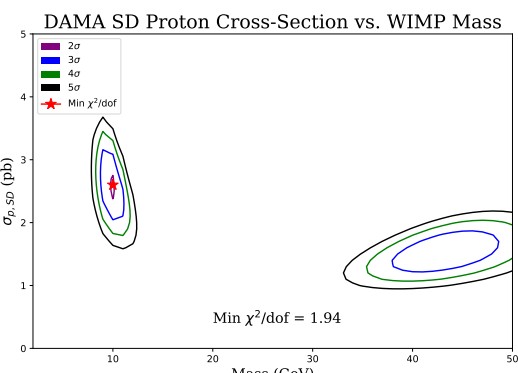

Figure 1: Best-fit parameter regions for the DAMA annual modulation in terms of spin-dependent scattering on neutrons (left) and protons (right) when considering a threshold of 0.75 keVee [10]. This plot has been produced using a new version of the public code DDCALC [16–18], which will be released in the near future, assuming standard quenching factors ($Q_{Na} = 0.3$ and $Q_I = 0.09$) and the energy resolution from Ref. [19].

but the more model-independent search for annual modulations is currently compatible both with DAMA and with no modulation [21]. These constraints will however improve considerably with further data-taking, as well as with the planned upgrade COSINE-200, which is anticipated to achieve lower background and threshold.

The second NaI experiment to test the DAMA claim is ANAIS-112 at Canfranc Underground Laboratory in Spain, which has achieved a 1 keV analysis threshold. Although the background level is slightly higher than DAMA, its time dependence is well understood and there is no evidence for annual modulations. Based on the current exposure from three years of data taking, ANAIS-112 and DAMA are incompatible at more than $3\sigma$ [22]. The significance of the exclusion is expected to increase to over $4\sigma$ with additional data taking and improved background rejection.

Additional experiments that aim for a particularly high radio purity and expect to be able to independently test the DAMA claim in the near future are PICOLON [23] and SABRE [24]. The latter is particularly innovative in that it aims for two different sites in the two hemispheres: one in the Gran Sasso National Laboratory in Italy and one in the Stawell Underground Physics Laboratory in Australia.

A novel strategy to probe the DAMA modulation has recently been proposed by the COSI-NUS collaboration [25]. The idea is to operate NaI crystals as low-temperature calorimeters, using the phonon channel to achieve much higher energy resolution and much lower threshold than otherwise possible. The simultaneous observation of heat and scintillation light furthermore allows for the discrimination of electron and nuclear recoils and a correspondingly strong background suppression. The potentially background-free environment makes it possible to test DAMA within a single annual cycle, using the fact that the modulation amplitude cannot be larger than the average absolute rate [26]. The required target mass of around 1 kg is expected to be available for data taking in 2023.

If these experiments, as widely assumed, yield null results, it will be essential to understand whether there exists any residual model dependence in the comparison with DAMA. In this context, it will be essential to establish whether the quenching factors relating the observed and true recoil energy may vary from detector to detector. If for example it turns out that these quenching factors depend on the growth method of the crystal or the concentration of Tl doping, the energy range probed by different experiments would differ and the comparison

would once again become model-dependent. Accurate measurements of the NaI quenching factors, as carried out for example in Ref. [27], are therefore an integral part of every effort to test the DAMA modulation.

If, on the other hand, the DAMA signal is confirmed, we need to address awkward questions about our understanding of DM. Indeed, after a considerable effort by the entire community, there is not a single consistent model that would give a good fit to the DAMA modulation while evading all other constraints. A convincing signal in NaI detectors would therefore force us to fundamentally re-think the interactions between dark and visible matter and the distribution of DM in the Milky Way.

## 4  Inelastic dark matter

To illustrate the amount of innovation and creativity in the community trying to find viable models to explain the DAMA anomaly, I would like to review a particularly interesting idea, which has proven ultimately unsuccessful in explaining DAMA but has led to a variety of other activities. The idea is to consider a fermionic DM particle with a Dirac mass term $m_D$ and introduce an additional Majorana mass term $m_M$ through spontaneous symmetry breaking.[2] For $m_M \ll m_D$ this leads to two nearly degenerate mass eigenstates $\chi$ and $\chi^*$ with mass splitting $\delta$. Interestingly, the mass eigenstates turn out to have off-diagonal couplings, i.e. all interactions must involve either the transition $\chi \to \chi^*$ or $\chi^* \to \chi$. Because of the energy required to overcome the mass splitting, this model has been named "Inelastic DM" (IDM) [29].

In IDM models, there may be a population of excited states, produced either in the early Universe (if the decay into the ground state is sufficiently slow) or produced via upscattering on cosmic rays, the Sun or the Earth [30]. These excited states can give rise to qualitatively new signals in direct detection experiments if they either de-excite spontaneously in the detector ($\chi^* \to \chi + \gamma$, called Luminous DM [31]) or if they release the energy $\delta$ upon scattering off nuclei or electrons ($\chi^* + X \to \chi + X$, called Exothermic DM [32]). As pointed out in Ref. [33] the resulting event rate may depend on the orientation of the detector relative to the velocity of the Earth through the Milky Way, leading to a characteristic daily modulation of the signal, which offers another way of distinguishing signal from background.

Alternatively, one can look for the production and subsequent decays of $\chi^*$ at colliders. This process is particularly interesting if the excited state is sufficiently long-lived that its decay gives rise to a displaced vertex. Ref. [34] proposed a strategy to search for such events at Belle II. The specific signature depends on whether DM is produced in isolation or in association with the dark Higgs boson responsible for generating the Majorana mass term [35]. Indeed, such a dark Higgs boson may also be long-lived, such that one can search for two displaced vertices involving each a pair of charged leptons or charged mesons. An official search for this signature by the Belle II collaboration is ongoing, so that exciting results may be expected in the near future.

## 5  Conclusion

Annually modulating event rates are among the most striking predictions for DM direct detection experiments and have triggered large experimental efforts. After more than a decade of data taking, the DAMA annual modulation keeps growing in significance and has now been observed for the first time for energies below 1 keVee. Although independent analyses of the DAMA signal are complicated by the lack of publicly available data, it seems clear that there is

---

[2]For an alternative construction, see Ref. [28].

no consistent interpretation of all direct detection experiments in terms of any known model of DM-nucleus scattering.

It is therefore essential to perform a completely model-independent test of the DAMA signal using independent NaI experiments. Two such experiments, namely COSINE-100 and ANAIS-112 have already published results that are in tension with the DAMA signal and are expected to become even more constraining in the near future. Further insights are expected from the upcoming ultra-pure detectors PICOLON and SABRE, the latter aiming for one detector each on both hemispheres. Finally, the COSINUS collaboration is pursuing the innovative idea to operate NaI crystals as low-temperature calorimeters.

Given these developments, the next 2–3 years will be decisive for our understanding of DM annual modulations. Clearly, a confirmation of the DAMA anomaly would be groundbreaking, but its implications for DM research and model-building are completely unclear. Nevertheless, given the creativity that the community has exhibited in the past, we can certainly look forward to new DM models being proposed, which will in turn inspire novel search strategies.

# Acknowledgements

I would like to thank Govinda Adhikari, Ken-Ichi Fushimi, Ambra Mariani, Florian Reindl, Marisa Sarsa, Karoline Schaeffner and Madeleine Zurowski for providing material for this presentation and Jonathan Cornell and Masen Pitts for coding up the most recent DAMA results in DDCALC.

**Funding information** FK acknowledges funding from the Deutsche Forschungsgemeinschaft (DFG) through the Emmy Noether Grant No. KA 4662/1-1.

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
