# Peer review of "Annual modulation of dark matter signals: Experimental results and new ideas"

_SciPost Physics Proceedings, doi:SciPost Phys. Proc. 12, 004 (2023)_

## Round 1 · Referee Report · Anonymous (Referee 1) · 2022-10-10

Report

The paper, contribution to the Proceedings of the IDM 2022 conference, reviews about the annual modulation of the dark matter signals in NaI(Tl), both in terms of experimental and model-building efforts. The paper can be published, once the small remarks in the following will be fulfilled.

- page 1 it is written: “… is predicted to switch sign at very low recoil energies (so-called anti-modulation)…”. Actually, once considering the energy resolution of the detectors such a “anti-mod” can be smeared out.

- page 1, the footnote: The author cites the paper [7]; however, the arguments of [7] have already been confuted by DAMA in ref:
1) Prog. Part. Nucl. Phys. 114 (2020) 103810 and
2) arXiv:2209.00882 (your ref. [10])
For an unbiased view, these should be cited as well.

- page 2: “… the DAMA collaboration has been observing an annual modulation in their nuclear recoil data…”. Actually, ... in the experimental single-hit data. No selection between electromagnetic and recoil data can be done at low energy.

- page 2: it is cited ref [8] for DAMA, however more updated ref is Nucl. Phys. At. Energy 22 (2021) 329 and your ref. [10]. The significance is slightly larger.

  • validity: -
  • significance: -
  • originality: -
  • clarity: -
  • formatting: -
  • grammar: -

Author:  Felix Kahlhoefer  on 2022-11-28  [id 3081]

(in reply to Report 1 on 2022-10-10)
Category:
reply to objection

I am grateful for the various suggestions. I detail below how they have been addressed in the resubmission:

  • page 1 it is written: “… is predicted to switch sign at very low recoil energies (so-called anti-modulation)…”. Actually, once considering the energy resolution of the detectors such a “anti-mod” can be smeared out.

Page 1 discusses the physical recoil spectrum, not the observed one, so the statement about anti-modulation is correct. However, I have added a comment on the possible effect of energy resolution in the context of DAMA on page 2.

  • page 1, the footnote: The author cites the paper [7]; however, the arguments of [7] have already been confuted by DAMA in ref: 1) Prog. Part. Nucl. Phys. 114 (2020) 103810 and 2) arXiv:2209.00882 (your ref. [10]) For an unbiased view, these should be cited as well.

The footnote on page 1 makes a very general statement without any reference to the DAMA experiment (which at this point in the text has not even been introduced yet). I believe the statement in the footnote to be factually correct, and no implication regarding DAMA is being made. I therefore find it unnecessary to add further references.

  • page 2: “… the DAMA collaboration has been observing an annual modulation in their nuclear recoil data…”. Actually, ... in the experimental single-hit data. No selection between electromagnetic and recoil data can be done at low energy.

I agree and have changed the text accordingly.

  • page 2: it is cited ref [8] for DAMA, however more updated ref is Nucl. Phys. At. Energy 22 (2021) 329 and your ref. [10]. The significance is slightly larger.

I agree and have changed the text accordingly.

Author:  Felix Kahlhoefer  on 2022-11-28  [id 3080]

(in reply to Report 1 on 2022-10-10)
Category:
reply to objection

I am grateful for the various suggestions. I detail below how they have been addressed in the resubmission:

  • page 1 it is written: “… is predicted to switch sign at very low recoil energies (so-called anti-modulation)…”. Actually, once considering the energy resolution of the detectors such a “anti-mod” can be smeared out.

Page 1 discusses the physical recoil spectrum, not the observed one, so the statement about anti-modulation is correct. However, I have added a comment on the possible effect of energy resolution in the context of DAMA on page 2.

  • page 1, the footnote: The author cites the paper [7]; however, the arguments of [7] have already been confuted by DAMA in ref: 1) Prog. Part. Nucl. Phys. 114 (2020) 103810 and 2) arXiv:2209.00882 (your ref. [10]) For an unbiased view, these should be cited as well.

The footnote on page 1 makes a very general statement without any reference to the DAMA experiment (which at this point in the text has not even been introduced yet). I believe the statement in the footnote to be factually correct, and no implication regarding DAMA is being made. I therefore find it unnecessary to add further references.

  • page 2: “… the DAMA collaboration has been observing an annual modulation in their nuclear recoil data…”. Actually, ... in the experimental single-hit data. No selection between electromagnetic and recoil data can be done at low energy.

I agree and have changed the text accordingly.

  • page 2: it is cited ref [8] for DAMA, however more updated ref is Nucl. Phys. At. Energy 22 (2021) 329 and your ref. [10]. The significance is slightly larger.

I agree and have changed the text accordingly.

Anonymous on 2022-11-30  [id 3091]

(in reply to Felix Kahlhoefer on 2022-11-28 [id 3080])
Category:
suggestion for further work

Dear Author,

please resubmit an updated version for the revision.

---

## Round 2 · Referee Report · Anonymous (Referee 1) · 2022-12-12

Report

The paper can be published as it is now

---

## Round 2 · Author Response

I am grateful for the various suggestions. I detail below how they have been addressed in the resubmission:

---

## Round 2 · List of Changes

page 1 it is written: “… is predicted to switch sign at very low recoil energies (so-called anti-modulation)…”. Actually, once considering the energy resolution of the detectors such a “anti-mod” can be smeared out.
Page 1 discusses the physical recoil spectrum, not the observed one, so the statement about anti-modulation is correct. However, I have added a comment on the possible effect of energy resolution in the context of DAMA on page 2.

page 1, the footnote: The author cites the paper [7]; however, the arguments of [7] have already been confuted by DAMA in ref: 1) Prog. Part. Nucl. Phys. 114 (2020) 103810 and 2) arXiv:2209.00882 (your ref. [10]) For an unbiased view, these should be cited as well.
The footnote on page 1 makes a very general statement without any reference to the DAMA experiment (which at this point in the text has not even been introduced yet). I believe the statement in the footnote to be factually correct, and no implication regarding DAMA is being made. I therefore find it unnecessary to add further references.

page 2: “… the DAMA collaboration has been observing an annual modulation in their nuclear recoil data…”. Actually, ... in the experimental single-hit data. No selection between electromagnetic and recoil data can be done at low energy.
I agree and have changed the text accordingly.

page 2: it is cited ref [8] for DAMA, however more updated ref is Nucl. Phys. At. Energy 22 (2021) 329 and your ref. [10]. The significance is slightly larger.
I agree and have changed the text accordingly.

---

## Editorial Decision

published